# Identifying critically ill children at high risk of acute kidney injury and renal replacement therapy

Rachel J. McGalliard[1,2], Stephen J. McWilliam[1,3,4,5]*, Samuel Maguire[6], Caroline A. Jones[1], Rebecca J. Jennings[1], Sarah Siner[1], Paul Newland[1], Matthew Peak[1,7], Christine Chesters[1], Graham Jeffers[7], Caroline Broughton[2], Lynsey McColl[8], Steven Lane[7], Stephane Paulus[1,2], Nigel A. Cunliffe[1,2,5], Paul Baines[1], Enitan D. Carrol[1,2,5]

1 Alder Hey Children's NHS Foundation Trust, Liverpool, United Kingdom, 2 Institute of Infection, Veterinary & Ecological Sciences, University of Liverpool, Liverpool, United Kingdom, 3 Department of Women's and Children's Health, Institute of Life Course & Medical Sciences, University of Liverpool, Liverpool, United Kingdom, 4 MRC Centre for Drug Safety Science, Institute of Systems, Molecular & Integrative Biology, University of Liverpool, Liverpool, United Kingdom, 5 Liverpool Health Partners, First Floor Liverpool Science Park, Liverpool, United Kingdom, 6 Medical School, University of Liverpool, Liverpool, United Kingdom, 7 Institute of Life Course & Medical Sciences, University of Liverpool, Liverpool, United Kingdom, 8 Select Statistical Services, Exeter, United Kingdom

* stevemcw@liverpool.ac.uk

**Data Availability Statement:** Data can be made available upon reasonable request to the University of Liverpool, Institute of Infection, Veterinary and Ecological Sciences Head of Operations

## Abstract

Acute kidney injury (AKI), a common complication in paediatric intensive care units (PICU), is associated with increased morbidity and mortality. In this single centre, prospective, observational cohort study, neutrophil gelatinase-associated lipocalin in urine (uNGAL) and plasma (pNGAL) and renal angina index (RAI), and combinations of these markers, were assessed for their ability to predict severe (stage 2 or 3) AKI in children and young people admitted to PICU. In PICU children and young people had initial and serial uNGAL and pNGAL measurements, RAI calculation on day 1, and collection of clinical data, including serum creatinine measurements. Primary outcomes were severe AKI and renal replacement therapy (RRT). Secondary outcomes were length of stay, hospital acquired infection and mortality. The area under the Receiver Operating Characteristic (ROC) curves and Youden index was used to determine biomarker performance and identify optimum cut-off values. Of 657 children recruited, 104 met criteria for severe AKI (15·8%) and 47 (7·2%) required RRT. Severe AKI was associated with increased length of stay, hospital acquired infection, and mortality. The area under the curve (AUC) for severe AKI prediction for Day 1 uNGAL, Day 1 pNGAL and RAI were 0.75 (95% Confidence Interval [CI] 0·69, 0·81), 0·64 (95% CI 0·56, 0·72), and 0.73 (95% CI 0·65, 0·80) respectively. The optimal combination of measures was RAI and day 1 uNGAL, giving an AUC of 0·80 for severe AKI prediction (95% CI 0·71, 0·88). In this heterogenous PICU cohort, urine or plasma NGAL in isolation had poorer prediction accuracy for severe AKI than in previously reported homogeneous populations. However, when combined together with RAI, they produced good prediction for severe AKI.

(iveshoo@liverpool.ac.uk). This request will need to include a formal request for collaboration, a study protocol, and clearly outlined study objectives. The decision to share data will lie with the University of Liverpool. A formal data sharing agreement will need to be agreed and signed prior to any (de-identified) data being exchanged.

**Funding:** This work was funded by the NIHR Biomedical Research Centre in Microbial Diseases, Liverpool and the Alder Hey Charity. The funders had no role in study design, data collection and analysis, decision to publish, or preparation of the manuscript.

**Competing interests:** The authors have declared that no competing interests exist.

## Introduction

Acute kidney injury (AKI) is a serious and common complication in critical care and is identified by abrupt (<48 hours) increase in serum creatinine levels or reduction in urine output resulting from injury, or insult causing a functional or structural change in the kidney [1]. The AWARE study, a multi-centre prospective study in paediatric critical care, demonstrated a 11·6% incidence of severe AKI and an absolute increase in mortality of 8·5% for those patients [2]. Severe AKI also confers an increase in morbidity in children through increased length of stay and subsequent chronic kidney disease [3]. Renal replacement therapy (RRT) may be required to maintain electrolyte, acid-base, and fluid balance. Timing of RRT is remains controversial, but studies in children suggest earlier implementation improves outcomes [4]. Accurate biomarkers would help clinicians to make a diagnosis earlier of AKI and thus weigh the risks and benefits of RRT for the individual based on objective evidence.

The diagnosis of AKI using the Kidney Disease: Improving Global Outcomes (KDIGO) is validated in children [5, 6]. However, this relies on urine output and changes in creatinine measurements that occur in the late stages of acute kidney injury, and are markers of established damage as opposed to reversible injury. This impedes early detection and instigation of preventative measures. Moreover, there is a lag time of several hours from injury to rise in creatinine that results in a 'sub-clinical' phase of AKI, which may be more amenable to treatment [7]. Lack of progress in reducing mortality in AKI has been attributed to delays in diagnosis [8]. These concerns are supported by evidence from the AWARE study, that serum creatinine measurements alone missed 67·2% of diagnoses of AKI, compared to urine output. However, in clinical settings, urine output measurements are variably recorded or monitored on wards and are dependent on favourable nursing staff ratios. Two suggested approaches to improve the early detection of AKI are the measurement of novel renal biomarkers, and use of a Renal Angina Index (RAI).

The ideal biomarker is outlined by the Wilson and Junger criteria, that identifies a pertinent condition during an early stage at which time effective, acceptable treatment can be instigated. A more sensitive, rapid, and renal related biomarker would allow further research into preventative strategies for AKI development and allow prompt initiation of personalised therapy to improve outcomes. Neutrophil Gelatinase Associated Lipocalin (NGAL) is a promising plasma and urinary biomarker that rises in the initial phase of AKI, but there is a wide variation of reported diagnostic accuracy in paediatric studies [9, 10]. Under conditions of stress and during proximal tubule insults, renal tubule epithelial cells secrete monomeric NGAL into the urine , whilst activated neutrophils release homodimers of NGAL [11].

The RAI has been developed as a screening tool that is validated in children to detect those at risk of AKI in intensive care units (ICU) and is calculated according to Table 1 [12]. In one study the RAI outperformed renal biomarkers for the prediction of severe AKI, and the combination of RAI with biomarker measurements further improved prediction [13]. This study aimed to evaluate the diagnostic and prognostic value of initial and serial urine (uNGAL),

**Table 1. Parameters to calculate renal angina index (adapted from Basu, 2017) [12].**

| Risk | | Injury: Serum Creatinine to baseline | | Renal Angina Index |
|---|---|---|---|---|
| ICU admission | 1 | No change | 1 | |
| Stem cell/ solid organ transplant | 3 | x 1–1·49 | 2 | Risk x Injury = (1–40) |
| Mechanical ventilation and/ or inotrope use | 5 | x 1·5–1·99 | 4 | Positive if ≥8 |
| | | x ≥2 | 8 | |

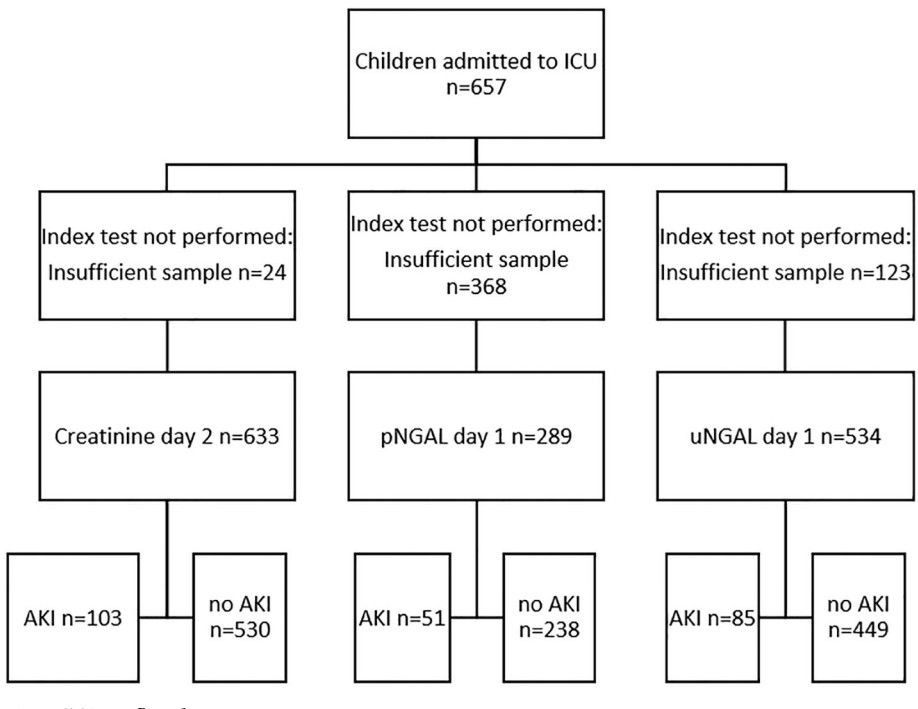

**Fig 1. STARD flowchart.**

plasma NGAL (pNGAL) concentrations, and the RAI to predict severe (stage 2 or 3) AKI and RRT in a heterogenous cohort of critically ill children.

## Materials and methods

### Study population

A prospective, observational cohort study of 657 children aged from 0–16 years, consecutively admitted to a tertiary paediatric ICU (PICU), were recruited from October 2010 to June 2012 (Fig 1). Exclusion criteria were; preterm infant age <37 weeks corrected gestational age or ≥16 years of age; children admitted moribund and not expected to survive more than 24 hours; children who were non-intubated elective admissions with a predicted duration of stay of less than 24 hours; children not expected to survive at least 28 days because of pre-existing condition; presence of existing directive to withhold life-sustaining treatment; end-stage renal disease requiring chronic dialysis therapy; and end-stage cirrhosis with evidence of portal hypertension. Ethics approval was granted by the local ethics committee (NRES Committee North West—Greater Manchester West, REC reference: 10/H1014/52) and all procedures followed were in accordance with the Helsinki Declaration. Written, informed consent was gained from the primary care giver.

### Sample and data collection and analysis

Patient demographic and clinical data were collected prospectively. Plasma urea, creatinine, uNGAL, and pNGAL (pNGAL was only measured on 295 children) were determined daily for the first 7 days of admission (see Fig 1 for STARD flowchart). Renal angina index was calculated retrospectively. Primary outcomes measures were incidence and risk factors for AKI, and RRT. Secondary outcomes were length of stay, hospital acquired infection, and mortality.

## Biomarker measurement

Serum creatinine was measured using an enzymatic (Creatininase/Creatinase) method developed by Abbott Diagnostics for use on the Abbott Architect Chemistry Analyser. uNGAL was measured using the ARCHITECT Urine NGAL assay (Abbott Diagnostics, Illinois, U.S.A.) and pNGAL was measured using a commercial ELISA kit (R&D Systems Inc., Minneapolis, USA), both according to the manufacturers' instructions.

## Definitions

Severe (stage 2 or 3) AKI within 72 hours of admission to ICU was defined using the serum creatinine criteria of the KDIGO guidelines [1]: a >2 times increase in the serum creatinine value from admission when compared to the value on day 3 (48–72 hours), or the need for renal replacement therapy. RRT was defined as the need for dialysis (haemodialysis or peritoneal dialysis), or haemofiltration. Baseline serum creatinine concentration was calculated as the lowest value in the preceding 3 months before admission to ICU if available. RAI was calculated retrospectively according to previously published protocols [14], using data obtained upon admission to PICU, and their first serum creatinine measurement on their first day in PICU.

A limiting factor in applying the RAI to this population was the number without baseline creatinine values. Calculation of baseline creatinine values using the Schwartz equation has been previously validated and widely used [2], but relies on height measurements which we did not have. We therefore conducted a secondary analysis using the Pottel method [15] to impute baseline creatinine values for those with absent data, following the method of Roy *et al* [16].

## Statistical analysis

Statistical analysis was performed using IBM SPSS Statistics version 22. Data were initially summarized using standard measures of location (mean, median etc.) and variation, counts and percentages are reported to summarize categorical variables. Chi-square test were used to compare categorical variables, Mann Whitney test for continuous variables. Receiver Operating Characteristic (ROC) curves were calculated by plotting sensitivity against 1-specificity for a range of cut-off values for each biomarker. The area under the curves (AUC) and Youden index was used to determine biomarker performance and identify optimum cut-off values. The closer the area under the ROC is to one the better the screening tool is at discriminating between positive and negative cases. Sensitivity and specificity were calculated for a range of cut-off values for uNGAL and pNGAL. Sensitivity and specificity curves were then plotted on the same graph and the point of crossover is the cut-off value that optimises sensitivity and specificity. This cut-off value was then used to classify patients as being positive (above the cut-off) or negative (equal to or below the cut-off). Multivariate logistic regression and Cox proportional-hazards models were fitted to the outcome renal replacement therapy, using the independent variables where an association was found in the univariate analysis.

## Results and discussion

### Study population

There were 2468 admissions to the PICU between October 2010 and June 2012. Nonetheless, 1339 were not eligible to participate and 472 were eligible but not recruited, thus 657 participants were enrolled in the study (Fig 2). There were 359 males (54·6%) and the median age at admission was 1·01 years (IQR, 0·30, 5·01) and the median weight on admission was 8·15kg

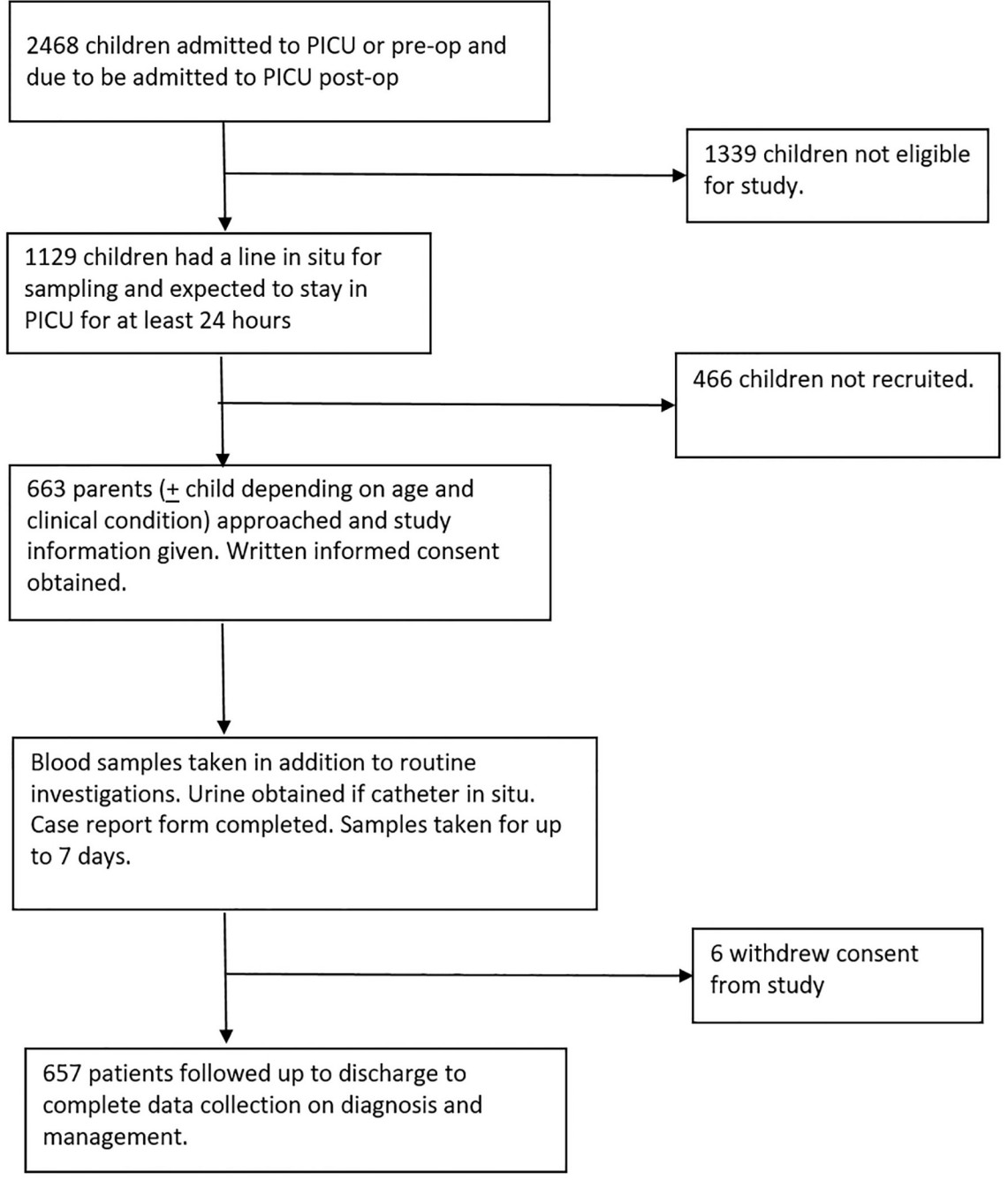

**Fig 2. CONSORT study inclusion and exclusion flowchart.**

(IQR, 4·15, 16·05). The most common reason for admission was for cardiac surgery (n = 350 53.3%) and suspected infection (n = 84 12·8%). The median duration of stay in the PICU for all patients was 2·92 days (IQR, 1·63, 6·00) and there were 15 deaths during the study (2·30%), with 12 being within the first 28 days (1·80%). Table 2 shows the characteristics of all patients stratified by the occurrence of severe AKI.

**Table 2. Clinical and demographic characteristics as stratified by AKI status.**

| Characteristic | Severe (stage 2 or 3) AKI | | P value |
|---|---|---|---|
| | **No n = 553 (84·17%)** | **Yes n = 104 (15·83%)** | |
| Age in years, median (IQR), | 1·03 (0·30, 5·17) | 0·94 (0·18, 4·42) | 0·761 |
| Range | 0·00–16·43 | 0·00–15·91 | |
| Gender—male, n (%) | 298 (53·9) | 61 (58·7) | 0·370 |
| Weight on Admission (kg), mean (SD) | 13·63 (±14·98) | 14·10 (±17·54) | 0·883 |
| Range | 1·09–103·00 | 2·90–108·00 | |
| Median (IQR) | 8·10 (4·20, 16·05) | 8·73 (3·90, 16·00) | |
| Reason for Admission—n (%) | | | |
| Cardiac Surgery | 287 (51·9) | 63 (60·6) | |
| Congenital Heart Disease | 13 (2·4) | 7 (6·7) | |
| SBI | 61 (11·0) | 10 (9·6) | |
| Infection | 73 (13·2) | 11 (10·6) | |
| Post-Op Other | 48 (8·7) | 3 (2·9) | |
| Trauma | 23 (4·2) | 1 (1·0) | |
| Other | 48 (8·7) | 9 (8·7) | |
| Surgery—Yes n (%) (can be more than one) | 391 (70·7) | 82 (788) | 0·09 |
| (can be more than one) | | | |
| Cardiac | 300 (76·7) | 68 (82·9) | |
| Neurosurgery | 24 (6·1) | 5 (6·1) | |
| General | 14 (3·6) | 1 (1·2) | |
| Orthopaedic | 7 (1·8) | 2 (2·4) | |
| Thoracic | 12 (3·1) | 1 (1·2) | |
| Abdominal | 35 (9·0) | 6 (14·6) | |
| Plastic | 4 (1·0) | 1 (1·2) | |
| ENT | 7 (1·8) | 1 (1·2) | |
| Other | 9 (2·3) | 2 (2·4) | |
| Renal Replacement Therapy—Yes n (%) | - | 47 (45·2) | |
| Peritoneal Dialysis | - | 36 (76·6) | |
| Haemofiltration | - | 10 (21·3) | |
| Haemodialysis | - | 1 (2·1) | |
| Duration of Renal Replacement Therapy (days) | | | |
| Mean (SD); range | - | 3·53 (±3·20); 0·21–15·34 | |
| Median (IQR) | - | 2·58 (1·21, 4·75) | |
| Duration of PICU stay (days) | | | <0·0001 |
| Mean (SD); range | 4·35 (±5·63); 0·25–52·96 | 8·78 (±9·83); 0·63–61·50 | |
| Median (IQR) | 2·75 (1·29, 4·96) | 6·40 (3·23, 9·98) | |
| MODS, n (%) | | | <0·0001 |
| No | 485 (87·7) | 48 (46·2) | |
| Yes | 68 (12·3) | 56 (53·8) | |
| PICU Mortality, n (%) | | | <0·0001 |
| Missing | 4 (0·7) | 0 (0·0) | |
| Alive | 542 (98·0) | 96 (92·3) | |
| Dead | 7 (1·3) | 8 (7·7) | |
| 28 Day mortality, n (%) | | | 0·358 |
| Alive | 542 (98·4) | 98 (97·0) | |
| Dead | 9 (1·6) | 3 (3·0) | |
| Maximum PELOD | | | <0·0001 |

*(Continued)*

**Table 2.** (Continued)

| Characteristic | Severe (stage 2 or 3) AKI | | P value |
|---|---|---|---|
| | **No n = 553 (84·17%)** | **Yes n = 104 (15·83%)** | |
| Mean (SD); range | 10·94 (±5·49); 0·00–31·00 | 18·55 (±6·61); 2·00–43·00 | |
| Median (IQR) | 11·00 (11·00, 12·00) | 21·00 (12·00, 22·00) | |
| Hospital Acquired Infection n (%) | | | <0·0001 |
| Yes | 138 (25·0) | 51 (49·0) | |
| No | 415 (75·0) | 53 (51·0) | |
| Serious Bacterial Infection n (%) | | | 0·797 |
| Yes | 107 (19·3) | 19 (18·3) | |
| No | 446 (80·7) | 85 (81·7) | |
| Renal angina index n (%) | | | <0·0001 |
| Positive (i.e ≥8) | 113 (36·6) | 35 (11·3) | |
| Negative | 156 (50·5) | 5 (1·6) | |

Abbreviations: SD = Standard Deviation, IQR = Interquartile Range, MODS = Multiple Organ Dysfunction Syndrome and PELOD = Paediatric Logistic Organ Dysfunction Score.

## AKI incidence and outcomes

Of the 657 patients, 104 (15·83%) met criteria for severe AKI within the first 72 hours of admission. Age, weight, gender, and reason for admission did not differ between patients with severe AKI and those without (Table 2). The secondary outcome measures were length of stay and mortality, which were both different between the two groups. Patients who developed severe AKI had a median length of stay in the PICU over twice as long as those who did not; 6·4 days (IQR; 3·23–9·98) versus, 2·75 days (IQR; 1·29–4·96) (p<0·0001). Mortality during admission in those who did develop severe AKI was higher than in those who did not develop severe AKI (7·7% versus 1·6% (p = <0·0001)). Furthermore, the incidence of hospital acquired infection was 49% of patients with severe AKI but only 25% of those without (p = <0·0001). Finally, the median maximum Paediatric Logistic Organ Dysfunction (PELOD) score was higher in those who developed severe AKI, when compared with those who did not (21 (IQR; 12–22) versus 11 (IQR; 11–12) (p<0·0001).

## AKI biomarker evaluation

uNGAL, pNGAL and creatinine were measured daily. Table 3 shows the Day 1 values and Fig 3 demonstrates the longitudinal profiles of the biomarkers between the severe AKI and non-AKI groups. Over the initial 3 days after admission, pNGAL concentrations were higher in the severe AKI population than those who did not develop severe AKI. However, uNGAL and creatinine concentrations remained higher in patients with severe AKI throughout the 7 days included in the study. Longitudinal biomarker values were compared for their diagnostic

**Table 3. Day 1 median biomarker concentrations categorized by AKI status.**

| Day 1 values | No severe AKI (553) | Severe AKI (104) | P value |
|---|---|---|---|
| | **Median (Interquartile range)** | | |
| uNGAL (ng/mL) | 30·9 (11·35, 127·5) | 186·0 (73·2, 785·1) | <0·0001 |
| pNGAL (ng/mL) | 120·1 (70·0, 245·7) | 174 (116·7, 441·8) | 0·013 |

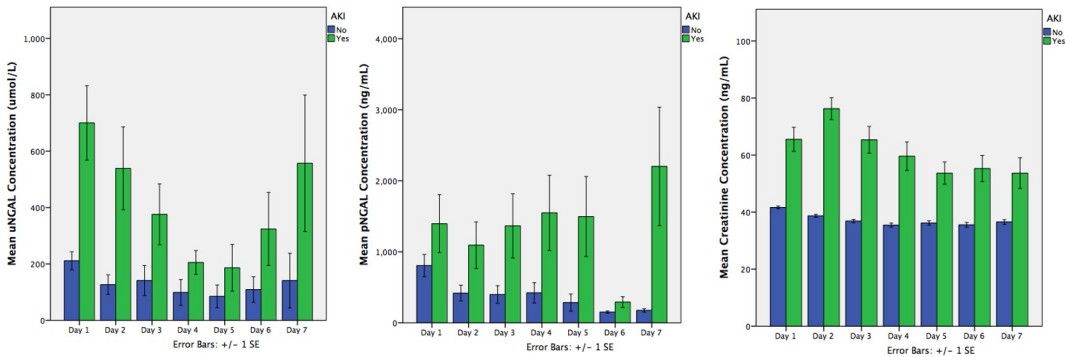

**Fig 3. Longitudinal profiles of mean biomarker levels over the first 7 days after PICU admission.** Blue bars no severe AKI, green bars severe AKI present.

accuracy using Receiver Operating Characteristic (ROC) curves (Table 4). The maximal AUC achieved was 0·75 (95% CI 0·69–0·81) for day 1 uNGAL with a cut off value of 61·15 ng/mL. pNGAL values showed poorer discriminative value over all time points.

## Renal angina index

Baseline creatinine measurements from the preceding 3 months were available in 309 patients (47%) and thus RAI was calculated using the formula in Table 1. RAI was defined as positive in 148 cases (i.e. score >7) with sensitivity of 0·88, specificity 0·58, positive predictive value 0·24 and negative predictive value 0·97 and AUC of the ROC curve of positive RAI for severe AKI was 0·73 (Table 4). Positive Likelihood Ratio was 2·08 and Negative Likelihood Ratio was 0·22.

Combining biomarker profiles (using the calculated cut-off values, via Youden's index to give binary high/ low values) with RAI, gave a maximal AUC of 0.80 (95% CI 0·71–0.88) for a combination of RAI and day 1 uNGAL (Table 4). This combination demonstrated increased specificity of 0.79 compared to either measure alone, but the sensitivity of 0.80 was lower than for RAI alone.

Using RAI or a biomarker value above the cut point (a positive result was returned if either test was positive, or both) led to increases in sensitivity and negative predictive value, with reduced specificity and positive predictive value. The maximal AUC achieved was 0.71 (95% CI 0.63–0.80) for a combination of RAI or day 2 uNGAL, with a sensitivity of 1.00, specificity of 0.43, positive predictive value of 0.28 and negative predictive value of 1.00 (Table 4).

Using the Pottel approach [15] to impute missing baseline creatinine values, RAI was positive in 352 of 657 cases with sensitivity of 0·85, specificity 0·52, positive predictive value 0·25 and negative predictive value 0·85 and AUC of the ROC curve of positive RAI for severe AKI was 0·80 (95% CI: 0.76–0.84).

## AKI following cardiac surgery

350 (53.3%) of the cohort were admitted to PICU following cardiac surgery. Of these 63 (18%) developed severe AKI. A secondary analysis of the predictive value of uNGAL, pNGAL and RAI for severe AKI was completed in this subgroup. Both uNGAL and pNGAL showed poorer diagnostic accuracy in this subgroup, than in the overall population (S1 and S2 Tables). Conversely, RAI demonstrated better diagnostic accuracy with an AUC of the ROC curve of 0.90 (95% CI: 0.86–0.94) (S2 Table).

**Table 4. AUC values for biomarkers of severe AKI.**

| Biomarker | AUC (95% CI) | Cut Point | Sensitivity | Specificity | PPV | NPV |
|---|---|---|---|---|---|---|
| Day 1 uNGAL | 0·75 (0·69, 0·81) | 61·15 | 0·80 | 0·62 | 0.29 | 0.94 |
| Day 2 uNGAL | 0·74 (0·67, 0·80) | 29·60 | 0·73 | 0·66 | 0.38 | 0.90 |
| Day 3 uNGAL | 0·66 (0·58, 0·73) | 30·70 | 0·63 | 0·62 | 0.42 | 0.80 |
| Day 1 pNGAL | 0·64 (0·56, 0·72) | 114·18 | 0·78 | 0·48 | 0.24 | 0.91 |
| Day 2 pNGAL | 0·68 (0·60, 0·77) | 123·63 | 0·78 | 0·55 | 0.31 | 0.90 |
| Day 3 pNGAL | 0·62 (0·52, 0·72) | 127·84 | 0·70 | 0·54 | 0.32 | 0.85 |
| RAI alone | 0·73 (0·65, 0·80) | - | 0·88 | 0·58 | 0.24 | 0.97 |
| RAI and day 1 uNGAL | 0.80 (0.71, 0.88) | - | 0.80 | 0.79 | 0.34 | 0.97 |
| RAI and day 2 uNGAL | 0.73 (0.61, 0.86) | - | 0.54 | 0.93 | 0.64 | 0.90 |
| RAI and day 1 pNGAL | 0.69 (0.56, 0.80) | - | 0.59 | 0.80 | 0.38 | 0.90 |
| RAI and day 2 pNGAL | 0.70 (0.56, 0.84) | - | 0.55 | 0.84 | 0.46 | 0.88 |
| RAI and day 1 uNGAL and day 1 pNGAL | 0.70 (0.54, 0.85) | - | 0.50 | 0.89 | 0.50 | 0.89 |
| RAI and day 2 uNGAL and day 2 pNGAL | 0.75 (0.59, 0.92) | - | 0.57 | 0.94 | 0.73 | 0.88 |
| RAI or day 1 uNGAL | 0.68 (0.59, 0.76) | - | 0.97 | 0.38 | 0.17 | 0.99 |
| RAI or day 2 uNGAL | 0.71 (0.63, 0.80) | - | 1.00 | 0.43 | 0.28 | 1.00 |
| RAI or day 1 pNGAL | 0.66 (0.56, 0.77) | - | 1.00 | 0.33 | 0.24 | 1.00 |
| RAI or day 2 pNGAL | 0.67 (0.56, 0.78) | - | 1.00 | 0.34 | 0.19 | 1.00 |
| RAI or day 1 uNGAL or day 1 pNGAL | 0.60 (0.48, 0.73) | - | 1.00 | 0.21 | 0.22 | 1.00 |
| RAI or day 2 uNGAL or day 2 pNGAL | 0.61 (0.46, 0.76) | - | 1.00 | 0.22 | 0.25 | 1.00 |

Cut point calculated using Youden's index.

## Renal replacement therapy incidence and outcomes

RRT was required in 47 (45.2%) patients with severe AKI. Median onset was 4 days after admission (IQR 2–8) and median duration of RRT was 2·58 days (IQR; 1·21–4·75). A total of 3 patients (6·4%) who required RRT died and 12 (25%) developed HAI, compared to 14 (2·3%) and 83 (13·6%) respectively, in the population who did not require RRT. The duration of ventilation and ICU stay was greater in the group undergoing renal replacement therapy 7·7 days

(IQR 5·1–11·8) and 9·2 days (IQR 6·9–20·8) which compared to those without RRT 2·8 days (IQR 1·4–5·2) and 3·0 days (IQR 1·7–6·3). A model for the prediction of RRT was developed using logistic regression and Cox proportional-hazards models and is presented in the S1 Appendix.

## Discussion

In our large, prospective single centre cohort of critically ill children, we demonstrate that severe AKI presents a significant disease burden in morbidity and mortality. The incidence of severe AKI was 15·8% with an absolute increase in mortality of 6·1%. Both severe AKI and RRT are associated with increased duration of PICU stay, and increased risk of HAI. Serial biomarker values are higher in children who develop severe AKI compared to those who do not. RAI, uNGAL or pNGAL alone all demonstrated moderate diagnostic accuracy for severe AKI. The combination of RAI with initial uNGAL values provided good prediction for severe AKI, and demonstrates the potential to improve the early identification of children who will develop severe AKI in in a general PICU population These approaches could help identify patients earlier that might benefit from RRT, and could potentially be used to define entry criteria for future clinical trials of pre-emptive RRT. Our study is the first to our knowledge that reports a significant increase in hospital acquired infections in the severe AKI population with a doubling of relative risk (25% to 49%).

Results from our study are comparable to the multicentre AWARE study which reported a 11·6% AKI incidence and an absolute increase in mortality of 8·5% (from 11% mortality in patients with AKI compared to 2·5% in the general cohort) [2]. In our study, severe AKI developed in 18% of children post-cardiac surgery, which is lower than other studies reporting rates between 25–42% [17, 18]. Other studies have shown the development of AKI whilst in PICU confers a 4-fold increase in total length of hospitalization, and in our cohort length of PICU stay was doubled in those with severe AKI [19].

In a recent systematic review and meta-analysis of NGAL evaluation in children by Filho et al, uNGAL demonstrated an AUC of 0·94 from 13 studies and for pNGAL an AUC of 0·90 [20]. However, in these studies NGAL measurements occurred 2–6 hours after surgery and predominantly in ICU admissions after cardiopulmonary bypass or contrast- induced nephropathy. In our clinically heterogenous population, day 1 uNGAL demonstrated only fair diagnostic accuracy (AUC = 0·75). In our post-cardiac surgery subgroup it performed worse, with an AUC of 0.53. Other less selective ICU studies have also found NGAL to be a poor biomarker of AKI and RRT in adult ICUs [21]. There are various possible explanations for this as timing of NGAL sampling is known to be critical in its diagnostic accuracy due to a rapid initial peak, and this may have been missed due to sampling occurring at variable times after ICU admission in our study where the initial peak may have been missed [22]. Secondly, heterogeneity of aetiological factors in renal insults may impact the utility of NGAL as a biomarker for general ICU populations. Thirdly, although the reference gold standard of paediatric AKI diagnosis uses KDIGO criteria including creatinine, it may not be adequately sensitive thus missing a proportion of diagnoses and may introduce diagnostic bias [2, 21].

We found that RAI in this heterogeneous cohort had moderate sensitivity and specificity, but excellent negative predictive value. Our findings are comparable to those in two previous publications which also demonstrated the high negative predictive value of RAI [13, 23]. In these studies, when RAI was combined with urinary NGAL values the AUC was improved to between 0·85 and 0·97 [13, 23]. In our study, the optimal combination was day 1 uNGAL and RAI, giving an AUC of 0·80. This approach improved specificity over either measure alone, and maintained a high negative predictive value.

Combining measures using 'or' (returning an overall positive result when any or all of the selected measures were positive) maximized sensitivity and negative predictive value at the expense of AUC, specificity and positive predictive value. For instance, the combination of RAI or day 2 uNGAL had a sensitivity and negative predictive value of 1.00. It may be important to consider such combinations in clinical practice as they would allow early exclusion of patients who will not develop AKI (based on a negative result for all measures) and closer monitoring of those with a positive result, confident that this group will include all patients who will develop AKI.

A total of 7·2% of children admitted to PICU required RRT. This is consistent with adult studies estimating RRT requirement in 5–10% of ICU patients [24]. A total of 45·2% of patients who developed severe AKI required RRT, predominantly via peritoneal dialysis. This is similar to an overall 2·9–13·4% incidence of RRT in a multicenter PICU study and other reports of 77–82% RRT initiation due to AKI related reasons in PICU [2, 25]. In our study, RRT was associated with increased incidence of adverse outcomes including absolute increases in duration of ventilation and ICU stay, 4·9 days and 6·2 days respectively. Studies from low resource settings reported mortality from RRT at 25–28%, which is higher than the 6.4% in our study. However, there is a paucity of general RRT related mortality in PICU [26].

Meta-analysis of RRT timing suggests differential efficacies of early RRT in adults compared to children [4, 27]. The literature suggests a survival benefit of early RRT in children with sepsis, AKI, and fluid overload [28]. This gap in clinical knowledge regarding optimal timing of RRT in PICU and accurate risk stratification, is an important area for further research [1, 29]. Given that there is currently no specific treatment to reverse AKI, it is vital that it is recognized promptly and managed aggressively to prevent further deterioration in renal function. More sensitive and rapid biomarkers, possibly combined with clinical risk scores such as the RAI could potentially be used to identify children who would benefit from early pre-emptive RRT.

The limitations of this study include the fact that this was a single centre study, and without data regarding hourly urine output some diagnoses of AKI may have been missed. Variation in times when blood and urine samples were collected after admission may also mean that peak concentrations were missed. Furthermore, specific indications for modality of RRT were not documented. RAI could not be calculated in 348 patients due to no previous creatinine measurement in the previous 3 months. This may have led to selection bias towards patients undergoing surgery with planned pre-operative bloods.

## Conclusions

This study supports the evidence base regarding the disease burden and adverse outcomes associated with severe AKI and RRT in a heterogeneous PICU population. The combination of RAI with initial uNGAL values provides good prediction for severe AKI in a general PICU population. Young age and cardiac surgery are associated with increased risk for severe AKI and RRT. The development of both severe AKI and RRT are associated with increased resource utilisation, namely PICU stay and nosocomial infections. Our RRT prediction model may allow further studies to stratify populations for early pre-emptive RRT timing in PICU.

## Supporting information

**S1 Table. Day 1 median biomarker concentrations categorized by AKI status in post-cardiac surgery subgroup.**
(DOCX)

**S2 Table. AUC values for biomarkers of severe AKI in post-cardiac surgery subgroup.**
(DOCX)

**S1 Appendix. Development of a model for prediction of renal replacement therapy.**
(DOCX)

## Acknowledgments

The authors would like to thank the children, young people, parents and carers who partici-
pated in the study. We also thank the PICU staff who contributed to the study.

## Author Contributions

**Conceptualization:** Matthew Peak, Stephane Paulus, Nigel A. Cunliffe, Paul Baines, Enitan D.
Carrol.

**Data curation:** Rebecca J. Jennings, Sarah Siner, Lynsey McColl, Steven Lane.

**Formal analysis:** Rachel J. McGalliard, Stephen J. McWilliam, Samuel Maguire, Christine
Chesters, Graham Jeffers, Caroline Broughton, Lynsey McColl, Steven Lane.

**Investigation:** Rebecca J. Jennings, Sarah Siner.

**Methodology:** Matthew Peak, Stephane Paulus, Nigel A. Cunliffe, Paul Baines, Enitan D.
Carrol.

**Project administration:** Rebecca J. Jennings, Sarah Siner.

**Writing – original draft:** Rachel J. McGalliard, Samuel Maguire.

**Writing – review & editing:** Stephen J. McWilliam, Caroline A. Jones, Paul Newland, Lynsey
McColl, Enitan D. Carrol.

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
