## [Decision Letter · Decision Letter 0]

12 Jun 2020

PONE-D-20-09458

Identifying critically ill children at high risk of acute kidney injury and renal replacement therapy

PLOS ONE

Dear Dr. McWilliam,

Thank you for submitting your manuscript to PLOS ONE. After careful consideration, we feel that it has merit but does not fully meet PLOS ONE’s publication criteria as it currently stands. Therefore, we invite you to submit a revised version of the manuscript that addresses the points raised during the review process.

We look forward to receiving your revised manuscript.

Kind regards,

Rajendra Bhimma, PhD

Academic Editor

PLOS ONE

Journal Requirements:

"Ethics approval was granted by the local UK research ethics committee (REC

10/H1014/52). Written, informed consent was gained from the primary care giver.

Once you have amended this statement in the Methods section of the manuscript, please add the same text to the “Ethics Statement” field of the submission form (via “Edit Submission”).

Additional Editor Comments (if provided):

Thank you very much for submitting your manuscript to PLOS ONE. Following a detailed review of the manuscript by three reviewers, there have been several comments raised by the reviewers that need to be addressed (see below). Please do this in accordance with the requirements of the journal.

Reviewers' comments:

Reviewer's Responses to Questions

**Comments to the Author**

1. Is the manuscript technically sound, and do the data support the conclusions?

Reviewer #1: Yes

Reviewer #2: Yes

Reviewer #3: No

2. Has the statistical analysis been performed appropriately and rigorously? 

Reviewer #1: Yes

Reviewer #2: Yes

Reviewer #3: No

3. Have the authors made all data underlying the findings in their manuscript fully available?

Reviewer #1: Yes

Reviewer #2: Yes

Reviewer #3: No

4. Is the manuscript presented in an intelligible fashion and written in standard English?

Reviewer #1: No

Reviewer #2: Yes

Reviewer #3: No

5. Review Comments to the Author

Reviewer #1: McGalliard and colleagues perform a prospective analysis of the ability of uNGAL and pNGAl to predict SCr based KDIGO Stage 2 AKI in critically ill children. They then assessed the ability of the RAI to do the same, and then combined the RAI and NGAL levels to determine if the performance improved. Then then undertook an extensive assessment of multiple factors to predict RRT provision in the cohort. The authors found that combination of the RAI and Day uNGAL provided the best predictive performance. The following issues require attention:

1) The section developing the model for RRT prediction can be shortened signficantly. I find it to be distracting from the overall message, and in fact, is not even mentioned in the abstract. As the authors note, RRT provision is a soft outcome measure, as indications for RRT provision were not provided in the protocol. I would recommend significant shorterning of this part of the manuscript, or removing it altogether.

2) There are two distinct cohorts in this population -- a group of heterogenous PICU patients and post-cardiac surgery patients. The RAI has never been validated in the post-cardiac surgery patients. It would be helpful for the authors to do a sensitivity analysis of RAI and NGAL performance in these two cohorts separately.

3) The authors define AKI as Stage 2 AKI--this should be called severe AKI or Stage 2 AKI throughout.

4) The authors need to state when the RAI was calculated. In the studies cited in there references, it was calcuated at 12 hours of PICU admission.

5) I am confused by the UOP criteria---were they used or not? In the Methods, the authros state they were (although the criteria they cite is Stage 1 AKI by KDIGO, this needs to be reconciled), but in the Discussion they state they don't have data on UOP? If they did use UOP, they need to use the correct KDIGO staging for Stage 2, and in the results, theshould note how manay patients had Stage 2 AKI by SCr, UOP or both.

6) A substantial percentage of patients were excluded because they did not have a reference SCr in the 3 months prior to PICU admission. Numerous pediatric studies, including AWARE have imputed baseline SCr in such cases based on the method of Zappitelli. The authors could also use the Pottel method. In any case, these patients should be assessed.

Minor

1) Data is the plural form of datum.

2) The authors do not need to state a difference is "significant", that is implied by the methods. If the AKI group has a higher value, then just state, "Patients with AKI has a longer duration of PICU stay", for example.

3) Please note the first mortality variable in the AKI comparision table is PICU mortality.

Reviewer #2: A question to the authors: What are the costs for the tests especially if they were point of care tests which could determine if the child was in established AKI and if so to commence renal replacement therapy sooner rater than later.i.e. would it be cost effective done for the right indication?

Reviewer #3: A prospective observational study was conducted to assess the ability of biomarkers, neutrophil gelatin-aseassociated lipocalin in urine (uNGAL) and plasma (pNGAL), and renal angina index (RAI) to predict acute kidney injury (AKI). The AUC of ROC curves were used to identify the best cut-points in biomarkers. The manuscript requires further clarification of the methods and results before drawing conclusions.

Major revision:

1- The manuscript lacks clarity and organization. The methods and results have not been clearly explained.

2- Clearly indicate how RAI and uNGAL/pNGAL values were used to calculate the sensitivity, specificity, PPV and NPV shown in table 4.

3- Further clarify the statistical analysis section by providing details of the receiver operating characteristic curve.

Minor revisions:

1- Abstract: More fully develop the abstract to help clarify the purpose, statistical methods and results of the study.

2- Line 135: Chi-square tests were used to COMPARE categorical variables.

3- Line 138: Clarify the sentence, “Logistic regression and Cox proportional-hazards models were fitted to the outcome renal replacement therapy, using the independent variables where an association was found in the univariate analysis.” Were MULTIVARIATE logistic and Cox proportional-hazard models fitted?

4- Check the line spacing of results in Table 1, specifically for Renal Replacement Therapy.

5- Table 5: Clarify if standard error refers to the standard error of the mean.

6. PLOS authors have the option to publish the peer review history of their article (what does this mean?). If published, this will include your full peer review and any attached files.

Reviewer #1: Yes: Stuart L Goldstein

Reviewer #2: No

Reviewer #3: No

---

## [Author Response · Author response to Decision Letter 0]

22 Jul 2020

Please see the 'Response to Reviewers' document for full details.

5. Review Comments to the Author

Reviewer #1: McGalliard and colleagues perform a prospective analysis of the ability of uNGAL and pNGAl to predict SCr based KDIGO Stage 2 AKI in critically ill children. They then assessed the ability of the RAI to do the same, and then combined the RAI and NGAL levels to determine if the performance improved. Then then undertook an extensive assessment of multiple factors to predict RRT provision in the cohort. The authors found that combination of the RAI and Day uNGAL provided the best predictive performance. The following issues require attention:

1) The section developing the model for RRT prediction can be shortened signficantly. I find it to be distracting from the overall message, and in fact, is not even mentioned in the abstract. As the authors note, RRT provision is a soft outcome measure, as indications for RRT provision were not provided in the protocol. I would recommend significant shorterning of this part of the manuscript, or removing it altogether.

The section of the Results describing the development of the RRT prediction model and associated tables and figures has been largely removed and will be provided as supplementary material. A paragraph describing the incidence and outcomes of RRT is retained in the main manuscript.

2) There are two distinct cohorts in this population -- a group of heterogenous PICU patients and post-cardiac surgery patients. The RAI has never been validated in the post-cardiac surgery patients. It would be helpful for the authors to do a sensitivity analysis of RAI and NGAL performance in these two cohorts separately.

Thank you for your comment. We have completed an additional analysis in the sub-population of 360 post-cardiac surgery patients. The results section has been updated to include these results.

3) The authors define AKI as Stage 2 AKI--this should be called severe AKI or Stage 2 AKI throughout.

This has been amended throughout.

4) The authors need to state when the RAI was calculated. In the studies cited in there references, it was calcuated at 12 hours of PICU admission.

The RAI was calculated retrospectively using data obtained upon admission to PICU, and their first serum creatinine measurement on their first day in PICU. The text has been updated to include this detail.

5) I am confused by the UOP criteria---were they used or not? In the Methods, the authros state they were (although the criteria they cite is Stage 1 AKI by KDIGO, this needs to be reconciled), but in the Discussion they state they don't have data on UOP? If they did use UOP, they need to use the correct KDIGO staging for Stage 2, and in the results, theshould note how manay patients had Stage 2 AKI by SCr, UOP or both.

Apologies for the confusion. Urine output data was not collected, and was therefore not used for defining AKI. The text has been amended to clarify this.

6) A substantial percentage of patients were excluded because they did not have a reference SCr in the 3 months prior to PICU admission. Numerous pediatric studies, including AWARE have imputed baseline SCr in such cases based on the method of Zappitelli. The authors could also use the Pottel method. In any case, these patients should be assessed.

We have added a paragraph to the Results section to address this issue. The Pottel method has been applied as height data were not collected. 

Minor

1) Data is the plural form of datum.

This has been amended throughout.

2) The authors do not need to state a difference is "significant", that is implied by the methods. If the AKI group has a higher value, then just state, "Patients with AKI has a longer duration of PICU stay", for example.

This has been amended throughout.

3) Please note the first mortality variable in the AKI comparision table is PICU mortality.

This has been amended.

Reviewer #2: A question to the authors: What are the costs for the tests especially if they were point of care tests which could determine if the child was in established AKI and if so to commence renal replacement therapy sooner rater than later.i.e. would it be cost effective done for the right indication?

Thank you for your question. We are not in a position to perform a health economic analysis of this approach at present. Costs for the ARCHITECT NGAL test used are available in a recent NICE publication at https://www.nice.org.uk/guidance/dg39/chapter/3-Evidence

Reviewer #3: A prospective observational study was conducted to assess the ability of biomarkers, neutrophil gelatin-aseassociated lipocalin in urine (uNGAL) and plasma (pNGAL), and renal angina index (RAI) to predict acute kidney injury (AKI). The AUC of ROC curves were used to identify the best cut-points in biomarkers. The manuscript requires further clarification of the methods and results before drawing conclusions.

Major revision:

1- The manuscript lacks clarity and organization. The methods and results have not been clearly explained.

Thank you for your comments. We have taken the opportunity to significantly review the manuscript, taking into account this and your other comments. We hope you will find the clarity and organisation of the manuscript improved.

2- Clearly indicate how RAI and uNGAL/pNGAL values were used to calculate the sensitivity, specificity, PPV and NPV shown in table 4.

Sensitivity and specificity was calculated for a range of cut-off values for uNGAL and pNGAL. Sensitivity and specificity curves were then plotted on the same graph and the point of crossover is the cut-off value that optimises sensitivity and specificity. This cut-off value was then used to classify patients as being positive above the cut-off or negative equal to or below the cut-off. The cut-offs, sensitivity etc. are reported in table 4. 

3- Further clarify the statistical analysis section by providing details of the receiver operating characteristic curve.

The ROC curve is calculated by plotting sensitivity against 1-specificity for a range of cut-off values. The closer the area under the ROC is to one the better the screening tool is at discriminating between positive and negative cases. 

Minor revisions:

1- Abstract: More fully develop the abstract to help clarify the purpose, statistical methods and results of the study.

The abstract has been reviewed in order to address these aspects.

2- Line 135: Chi-square tests were used to COMPARE categorical variables.

This has been amended.

3- Line 138: Clarify the sentence, “Logistic regression and Cox proportional-hazards models were fitted to the outcome renal replacement therapy, using the independent variables where an association was found in the univariate analysis.” Were MULTIVARIATE logistic and Cox proportional-hazard models fitted?

Multivariate logistic regression and Cox proportional-hazard models were fitted. 

4- Check the line spacing of results in Table 1, specifically for Renal Replacement Therapy.

This has been amended.

5- Table 5: Clarify if standard error refers to the standard error of the mean.

The standard errors do refer to the standard errors of the mean. The relevant sections (now moved to supplementary) have been updated to clarify this.

---

## [Decision Letter · Decision Letter 1]

25 Sep 2020

Identifying critically ill children at high risk of acute kidney injury and renal replacement therapy

PONE-D-20-09458R1

Dear Dr. McWilliam,

We’re pleased to inform you that your manuscript has been judged scientifically suitable for publication and will be formally accepted for publication once it meets all outstanding technical requirements.

Kind regards,

Brenda M. Morrow, PhD

Academic Editor

PLOS ONE

Additional Editor Comments (optional):

Reviewers' comments:

Reviewer's Responses to Questions

**Comments to the Author**

1. If the authors have adequately addressed your comments raised in a previous round of review and you feel that this manuscript is now acceptable for publication, you may indicate that here to bypass the “Comments to the Author” section, enter your conflict of interest statement in the “Confidential to Editor” section, and submit your "Accept" recommendation.

Reviewer #1: (No Response)

Reviewer #2: All comments have been addressed

Reviewer #3: All comments have been addressed

2. Is the manuscript technically sound, and do the data support the conclusions?

Reviewer #1: (No Response)

Reviewer #2: Yes

Reviewer #3: (No Response)

3. Has the statistical analysis been performed appropriately and rigorously? 

Reviewer #1: (No Response)

Reviewer #2: I Don't Know

Reviewer #3: (No Response)

4. Have the authors made all data underlying the findings in their manuscript fully available?

Reviewer #1: (No Response)

Reviewer #2: Yes

Reviewer #3: (No Response)

5. Is the manuscript presented in an intelligible fashion and written in standard English?

Reviewer #1: (No Response)

Reviewer #2: Yes

Reviewer #3: (No Response)

6. Review Comments to the Author

Reviewer #1: (No Response)

Reviewer #2: The revision reads much better. More logically set out. All the tables and flow charts are more easily interpreted. The combination of UGal and RAI having a 80% accuracy is better than no criteria but further research to tease out the variables in different causes of AKI and how they are assessed and compared is needed for prediction of AKI and the timing of commencement of renal replacement therapy e.g. post surgical cases compared to general cases.

Reviewer #3: (No Response)

7. PLOS authors have the option to publish the peer review history of their article (what does this mean?). If published, this will include your full peer review and any attached files.

Reviewer #1: No

Reviewer #2: No

Reviewer #3: No

---

## [Editor Report · Acceptance letter]

1 Oct 2020

PONE-D-20-09458R1 

Identifying critically ill children at high risk of acute kidney injury and renal replacement therapy 

Dear Dr. McWilliam:

I'm pleased to inform you that your manuscript has been deemed suitable for publication in PLOS ONE. Congratulations! Your manuscript is now with our production department. 

Kind regards, 

on behalf of

Professor Brenda M. Morrow 

Academic Editor

PLOS ONE